# Effect of Passivation Layer on the Thin Film Perovskite Random Lasers

**DOI:** 10.3390/ma13102322

**Published:** 2020-05-18

**Authors:** Subha Prakash Mallick, Yu-Heng Hong, Lih-Ren Chen, Tsung Sheng Kao, Tien-Chang Lu

**Affiliations:** Department of Photonics, College of Electrical and Computer Engineering, National Chiao Tung University, Hsinchu 30010, Taiwan; subhaprakash118.eic06g@nctu.edu.tw (S.P.M); enochhong@me.com (Y.-H.H.); albertchen.eo07g@nctu.edu.tw (L.-R.C.); tskao@g2.nctu.edu.tw (T.S.K.)

**Keywords:** perovskite, random lasers, passivation layer, multiple scattering effect, speckle free image projection

## Abstract

Novel functionalities of disorder-induced scattering effect in random lasers, attributed to low spatial coherence, draw remarkable attention in high-contrast to superior quality speckle-free imaging applications. This paper demonstrates perovskite-polystyrene (PS)-based random lasing action with robust optical performance at room temperature. Optical characterizations are carried out upon perovskite thin films addition with polystyrene of different mixing concentrations (wt.%). A low threshold lasing operation is achieved with an increasing concentration of polystyrene, accompanying a wavy surface texture with high surface roughness. The rough surface dominating multiple scattering effects leads to enhanced feedback efficiency. Moreover, this study also elucidates efficient fabrication process steps for the development of high quality and durable PS-based random lasers. With the advantages of reduced coherent artifacts and low spatial coherence, speckle free projection images of the USAF (U. S. Air Force MIL-STD-150A standard of 1951) resolution test chart are shown for different PS-based random lasers.

## 1. Introduction

Recent technological advancements in the research of lead halide perovskites have drawn remarkable attention in the context of optoelectronics device applications. Hybrid metal halide perovskites such as MAPbX_3_ (MA—methyl ammonium and X-Cl, Br, or I) manifest as direct bandgap materials with large absorption coefficient, slow Auger recombination rate, low defect densities, and high optical gain. Furthermore, solvent engineered metal halide perovskites are deemed to be novel and extended researches to photovoltaics, LEDs, and lasers [1,2,3,4,5,6,7,8,9]. As reported, organic-inorganic lead halide perovskite-based laser devices are achieved by employing nanowires and whispering gallery modes for feedback oscillation [10,11,12]. Some other researches demonstrated resonators in tapered fiber design and self-organized nanorods in a microcrystalline network. Patterned and flattened perovskite thin films incorporated into distributed feedback (DFB) lasers, vertical cavity surface emitting lasers, photonic crystal lasers, and optical cavities were also reported. All such laser devices need to follow proper nano fabrication and controlled growth processes [13,14,15,16,17]. Thanks to the high optical gain in MAPbX_3_ materials, only a thin film perovskite with proper scattering centers is able to achieve random lasing action without a complicated laser cavity fabrication process [18,19,20,21,22,23]. Also, the hybrid perovskite materials with Br content are suitable for high quality, and shape controlled single crystal formation. In addition, the solution processed organometal halide perovskite thin film of MAPbBr_3_ is cost effective and shows faster crystallization with well interconnected grain boundaries [24,25]. Comprehensive analyses and novel approaches to improve the lasing properties and ASE in lead halide perovskites, such as wavelength range engineering, threshold reduction, lasing cavity geometry, controlled growth process, and stability improvement, etc. have been illustrated [26]. 

Intensive researches in random lasers have revealed greater potential applications in imaging, illumination, biomedical applications, sensing, etc. In comparison to other lasers, fabrication processes of random lasers are much easier, cost effective, and facilitate large-scale production. As per our previous report, lasing action was achieved by the solution process of organic-inorganic metal halide embedded with crystalline nanostructures. However, the lasing characteristic was frequently altered upon a temperature induced phase transition [17]. Several studies demonstrate that chemically engineered antisolvent dripping process enhances the stability and material response of perovskite. An antisolvent dripping process was introduced to synthesize the perovskite thin film. In this process, it provides complete surface coverage and high surface roughness with enriched polycrystalline grains [27,28,29,30,31,32,33,34,35]. This study depicts the efficient solvent engineered fabrication process steps for the development of high quality [1] and durable perovskite-polystyrene (PS)-based random lasers with robust optical performances at room temperature. The lasing action is achieved with mechanism of optical feedback process in the gain medium via multiple scattering. Thus, the photon lifetime in the gain medium is enhanced. As a reason, the efficiency of light amplification process is improved and results in amplified stimulated emission. In our previous research by Wang et al. [1], which demonstrates a low threshold lasing operation in perovskite random lasers upon bending and surface deformation. In a corollary to that, this study demonstrates, a low threshold lasing operation upon an increment of polystyrene concentration (wt.%), which introduces a wavy surface texture and high surface roughness. The rough surface induces multiple scattering effects, hence leads to enhanced light-matter interaction efficiency. Furthermore, the significance of having a polystyrene layer on the top of perovskite thin film is to enhance optical feedback efficiency due to the photon recycling process. This is induced by the multiple reflection processes at the perovskite/polystyrene and polyimide/perovskite interfaces. Concomitant to this a great disparity between the refractive indices of perovskite (n ~ 2.5), polystyrene (n ~ 1.5), and polyimide (n ~ 1.5) results in tighter light confinement, efficient light scattering, and enhanced photon lifetime. Further, this interface engineering is indispensable for perovskite thin film. The hydrolysis and degradation of perovskite in a humid atmosphere is obvious. The moisture contamination to perovskite decomposes the CH_3_NH_3_PbBr_3_ compound into PbBr_2_ and CH_3_NH_3_Br, which further degrades into HBr and CH_3_NH_2_, and form pores. Thus, we introduced polystyrene as a passivation layer on perovskite thin film, which effectively improves the stability of perovskite thin film and reduces nonradiative recombination that leads to an increase in carrier lifetime and photoluminescence intensity, etc. [36,37,38]. Furthermore, these random lasers are analyzed for the effectiveness of the projection light sources. A comparative imaging analysis with coherent laser source (Nd: YAG) demonstrates that the low coherence perovskite-polystyrene (wt.%) based random lasers are efficient in speckle free image projection.

## 2. Experimental Section 

### 2.1. Fabrication Process of Random Lasers

The fabrication process of perovskite (CH_3_NH_3_PbBr_3_), and perovskite-polystyrene (wt.%) random lasers is shown in Figure 1. The methylammonium bromide (MABr) and lead bromide (PbBr_2_) powder were added into a solvent of engineered γ-butyrolactone: dimethyl sulfoxide (GBL: DMSO) with a mixing ratio of 7:3. The solution was then stirred to ensure complete dissolution, which further turned into a transparent solution of 1 M concentration. Afterward, the solution was coated onto the UV-O_3_ cleaned glass substrates followed by a two-step spin coating process for even distribution. These two-step spin coating process includes rotating speed of 1000 rpm for 10 s followed by 5000 rpm for 50 s. The highly flexible PI substrate was tapped onto a glass substrate to get rid of wrinkles during the spin coating process. Meanwhile, antisolvent toluene was dripped at a constant period in the second stage of spin coating process. At final process step, the sample was annealed at 100 °C for 30 min to vaporize the residual solvents as well as to transit from the intermediate solvate phase to perovskite thin film. Soon after the formation of perovskite thin film, different concentrations of polystyrene (wt.%) were deposited by the spin coating process with a rotating speed of 1000 rpm for 10 s. The respective polystyrene thin films of 10, 20, and 30 wt.% concentrations were formed after annealing it with 100 °C for 5 min.

### 2.2. Micro-Photoluminescence Measurement Setup (μ-PL)

The experimental setup for micro-photoluminescence (μ-PL) measurement is shown in Figure 2. The optical pumping system applies Nd: YVO_4_, 355 nm pulse laser as a pumping source operated under 1 kHz repetition rate and 0.5 ns pulse width. A 0.55 numerical aperture (NA) microscopic objective lens is placed between the sample and the laser to collect the light reflected from the sample surface and transmit the light into a charge-coupled device (CCD); thus, a clear view of the sample surface can be seen. The experiments were carried out under room temperature. The excitation area dependent photoluminescence (PL) signal as a function of pumping energy density was received by a UV optical fiber of 600 μm core normal to the sample surface and fed into a nitrogen-cooled spectrometer (Jobin-Yvon IHR 320 Spectrometer, Kyoto, Japan) having the spectral resolution about 0.2 nm, through a charge-coupled device.

### 2.3. Experimental Setup for USAF Resolution Test Chart Imaging

The experimental setup for the imaging USAF resolution test chart is shown in Figure 3. The Nd: YVO_4_, 355 nm pulse laser is applied as the pumping source onto the random laser sample. The excited random laser light emission is collected by the objective lens of N.A. of 0.45, and magnification of 20×. After passing through a diffuser, the laser is projecting onto an USAF resolution test chart. The transmitted light from the USAF test chart is then focused by an objective lens of N.A. of 0.7 with a magnification of 50×, thereafter imaged by a CCD camera.

## 3. Results and Discussion

### 3.1. Surface Texture Analysis

Planar view scanning electron microscope (SEM) images and atomic force microscope (AFM) surface morphologies of different samples are shown in Figure 4a,b, respectively. The disordered rugged surface morphology by randomly distributed perovskite grains is found over the substrate. Planar views of add-on polystyrene thin films with a concentration varied from 10 to 30 wt.% show a hard-sticky film with rugged inhomogeneous wavy surface texture with incremental surface roughness. Root mean square values of surface roughness measured by AFM measurements are 20 nm, 22.1 nm, 24.1 nm, and 25.3 nm for samples consisted of perovskite, perovskite-polystyrene at 10, 20, and 30 wt.% concentrations, respectively. The optical characteristics of the samples are investigated by measuring the light-light (L-L) curve and full width at half maximum (FWHM) of the emission spectra, as well as the U.S. Air Force (USAF) resolution test chart for the utility of the projection light source. The lasing characteristics of perovskite thin film with different surface textures induced by varying polystyrene concentration passivation layers are investigated in this study.

### 3.2. Lasing Performance Analysis

Figure 5 displays the emission spectra of perovskite random lasers as a function of pumping energy density, and their corresponding L-L curves together with FWHM of emission spectra are shown in Figure 6. The diameter of the pumping spot is approximately 83 ± 5 μm. In principle, the random lasing action is achieved by the dominant multiple scattering effect. In this study, the dominant multiple scattering effect is provided by the microscale rugged surface morphology induced by inhomogeneous crystallographic perovskite grains as shown in Figure 1. With the increase of pumping energy density, the “s” shape of the L-L curve is observed in perovskite-polystyrene (wt.%) random lasers, accompanying the abrupt change of spectra linewidth at the measured threshold energy density. As shown in Figure 5a, the emission spectra may refer to a photoluminescent (PL) mode and a random lasing (RL) mode, coexisting in the light emission performance of the perovskite thin films. The central wavelengths of the PL and RL modes are at around 535 nm and 546 nm, respectively. With an increasement of the pumping powers, the full width at half maximum (FWHM) of the PL emission peak is almost invariant at around 24 nm. In terms of the RL mode, the band width may be reduced to about 6.17 nm at the maximum pumping energy density of 5.5 mJ/cm^2^. The extracted peak width at different pumping densities is presented in Figure 6a. Furthermore, multiple emission peaks can be observed in the RL mode, which may be one of the significant features of the random lasing action [39]. The measured intensities of each emission peaks may change along with the pumping power variation. Regarding to the lasing performance of the bare perovskite thin films, the L-L curve does not show a clear “s” shape in the log-log scale, which indicates that the perovskite sample only reaches the amplified spontaneous emission (ASE) performance with a threshold of 3.1 mJ/cm^2^. Further, a clear inference can be drawn from perovskite ASE spectral response that, it is evident of less spikes and wide spectral linewidth due to negligible optical feedback in comparison to other perovskite-polystyrene random lasers. This is a clear signature of diffusive light propagation due to long scattering mean free path. The emission spectra of perovskite-polystyrene random lasers with 10, 20, and 30 wt.% concentrations are shown in Figure 5b–d. The distinct multiple lasing peaks of perovskite-polystyrene random lasers are observed at varied pumping power comparing to the perovskite sample shown in Figure 5a. This indicates strong photon localization effect induced by the rugged surface morphologies with wavy surface texture. Different mode spacing could be a result of a deformation in cavity length due to external stress acquired by the wavy polystyrene surface texture [40,41]. As shown in Figure 5, red-shifted lasing peaks with respect to photoluminescence (PL) peak at about 535 nm are spotted for different random lasers, which results from less band tail absorption loss, when the gain-loss competition is considered [42]. Thus, the addition of different concentrations of polystyrene as a passivation layer improves the disorder strength, and photon lifetime due to multireflection processes at the interfaces. This turns out to decrease the lasing threshold. As a result of this, the lowest threshold energy density of 2.4 mJ/cm^2^ is achieved in perovskite-polystyrene 30 wt.% based random laser as shown in Figure 6d. Similarly, Figure 6b–d depict improved lasing characteristics with lowering down the threshold energy density values as 2.9 mJ/cm^2^, and 2.6 mJ/cm^2^ for perovskite-polystyrene 10 wt.%, and perovskite-polystyrene 20 wt.% concentration-based random lasers respectively.

### 3.3. Optical Characteristics of Random Lasers and Speckle Free Imaging

To study the threshold characteristics of different random lasers, it is observed that threshold energy density decreases upon increase in excitation area. This can be comprehended by Raith and Apalkov statistical analysis [43,44,45,46] of threshold behavior using power-law.
(1)$$E_{th}\propto \;e^{\left[-{\left(\frac{\ln\left(\frac{A}{A_{0}}\right)}{G}\right)}^{1/\lambda }\right]}$$
where E_th_ is the threshold energy density, A is the excitation area, A_0_ is the two-dimensional area acquired by quasi mode, G is disorder strength of the medium, and λ depends on degree of disordered correlation. For simplicity, the coefficient λ→1 is set as the first degree of disordered correlation. Thus, Equation (1) reduces to $E_{th}\propto {\left(\frac{A}{A_{0}}\right)}^{\frac{-1}{G}}$ which facilitates the comparison of different random lasing systems. Figure 7a shows the excitation area dependent threshold variation, where the experimental data (dots) is fitted by theoretical calculation (lines). By using power-law estimation, the inverse of disorder strength (1/G) is fitted as 1.17, 1.13, and 0.93 for perovskite-polystyrene random laser with polystyrene concentration of 10, 20, and 30 wt.% concentrations, respectively. It is well evident that disorder strength (G) increases with an increase in polystyrene concentration. As a result, it is clear to testify that under the large excitation area, the increase of polystyrene concentrations exhibits higher light-matter interaction efficiency and a dominant multiple scattering effect. This is due to the presence of a wavy surface texture complemented by rugged inhomogeneous surface morphology. Figure 7b shows the image of the USAF (U. S. Air Force MIL-STD-150A standard of 1951) resolution test chart [47,48] under illumination of different light sources. It is well evident that the full field image projection of the USAF test chart fails under illumination of Nd: YAG coherent light source. The intense illumination source with high spatial coherence generates coherent image artifacts, hence leads to speckle or corrupt image formation. To mitigate such effects and to improve image quality, perovskite-polystyrene (wt.%) based random lasers with low spatial coherence are used to produce speckle free, full-field image projections of USAF resolution test chart. Results from the contrast to noise ratio, verify the improved image quality using different perovskite-polystyrene (wt.%) based random lasers. To evaluate the image quality numerically, the contrast to noise ratio (CNR) is applied by the calculation of $\frac{\left(I_{f}\right)-I_{b}}{\left(\sigma_{f}+\sigma_{b}\right)/2}$, where <I_f_ > and <I_b_> are average of foreground and background intensities, $\sigma_{f}$ and $\sigma_{b}$ are the standard deviations of foreground and background intensities. It can be clearly seen that the perovskite-polystyrene random laser with polystyrene concentration of 30 wt.% concentration demonstrates the best CNR values. Although the perovskite ASE manifests a relatively low coherent light source, the low spectra intensity more or less degrades the imaging quality in comparison to the random laser light source. Figure 7c demonstrates the durability test for different perovskite samples. It is evident that, with a 120 h experiment under ambient conditions, the random lasers show degradation and a drop in photoluminescence intensity. However, the higher slope of the exponential curves manifests an increased lifetime upon the addition of concentrations of polystyrene (wt.%) as a passivation layer on the top perovskite thin film. Upon addition of polystyrene (wt.%), it provides wavy surface texture with an increased surface roughness that induces dominant multiple scattering effects which accounts for low threshold operation. An increased number of scattering centers that exhibit higher radiation efficiency incorporated into a longer lifetime prevails in perovskite-polystyrene (wt.%) random lasers.

## 4. Conclusions

In this study, we have demonstrated solvent-engineered efficient fabrication process steps for the development of high quality and durable perovskite-polystyrene (PS)-based random lasers with robust optical performance at room temperature. The addition of different concentrations of polystyrene (wt.%) on perovskite thin films exhibit a long lifetime and low threshold random lasing operation as a reason of higher surface roughness induced dominant multiple scattering effect. In consequence of low spatial coherence and reduced coherent image artifacts, perovskite-polystyrene (PS) based random lasers manifest full-field speckle free image projections of USAF resolution test chart. From the comparative analyses among all random lasers, the perovskite-polystyrene (30 wt.%) based random laser demonstrates the lowest threshold operation with the best CNR values in speckle free imaging. With an advantage of speckle free imaging, perovskite-polystyrene (PS) based random lasers provides a greater potential application to optical coherence tomography, the formation of the colored image, and image processing, etc.

## Figures and Tables

**Figure 1 materials-13-02322-f001:**
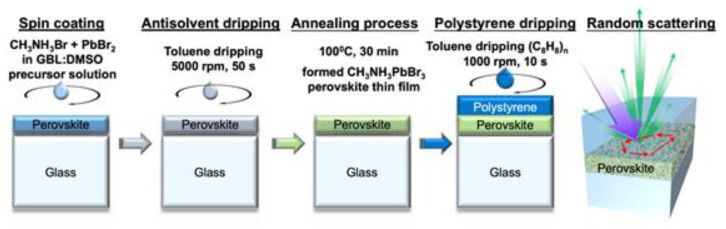
The schematic fabrication process flow for perovskite and polystyrene thin film formation. The random scattering effect is provided by the microscale rugged surface morphology of perovskite films.

**Figure 2 materials-13-02322-f002:**
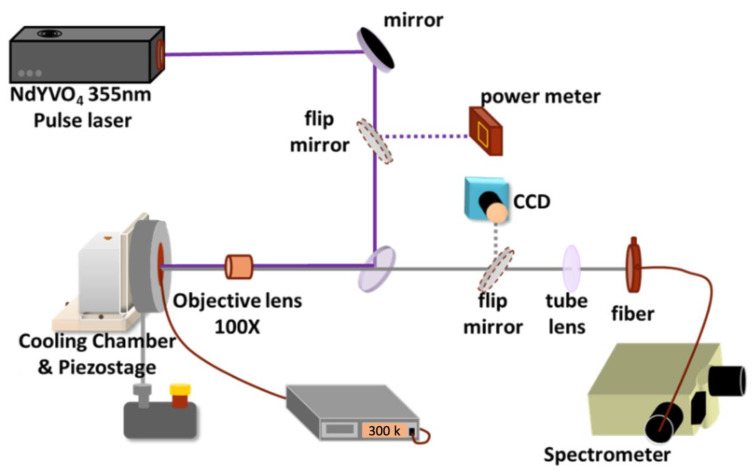
Experimental setup for Micro-Photoluminescence (μ-PL) measurement.

**Figure 3 materials-13-02322-f003:**
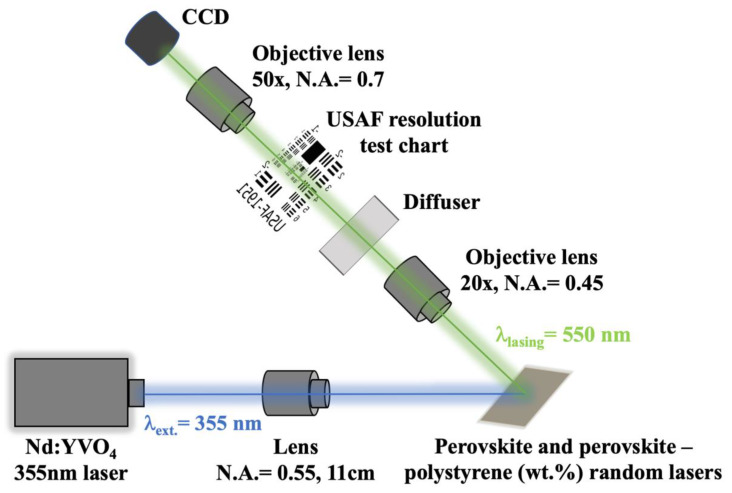
Experimental setup for USAF resolution test chart imaging.

**Figure 4 materials-13-02322-f004:**
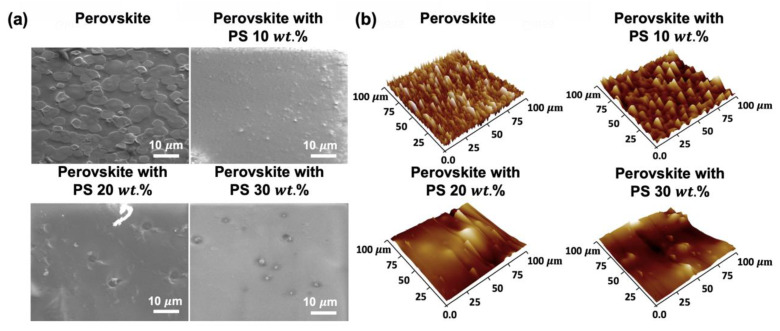
(**a**)**.** Planar SEM images and (**b**) AFM surface morphologies of different samples.

**Figure 5 materials-13-02322-f005:**
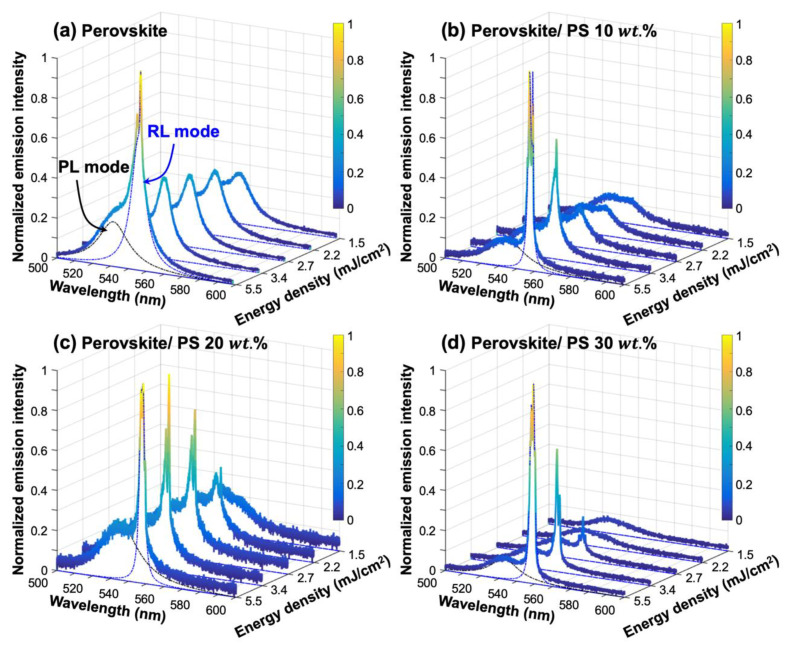
Spectral characteristics of different random lasers as a function of pumping energy density of: (**a**) perovskite sample; (**b**) perovskite-polystyrene 10 wt.% sample; (**c**) perovskite-polystyrene 20 wt.% sample; (**d**) perovskite-polystyrene 30 wt.% sample.

**Figure 6 materials-13-02322-f006:**
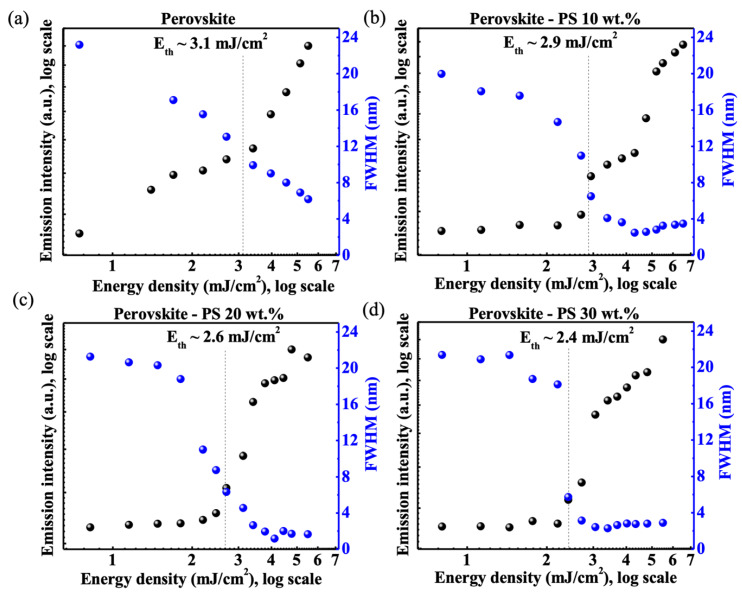
L-L-FWHM curve of different random lasers. The L-L-FWHM measurement results of (**a**) perovskite sample; (**b**) perovskite-polystyrene 10 wt.% sample; (**c**) perovskite-polystyrene 20 wt.% sample; (**d**) perovskite-polystyrene 30 wt.% sample at different pumping energy densities.

**Figure 7 materials-13-02322-f007:**
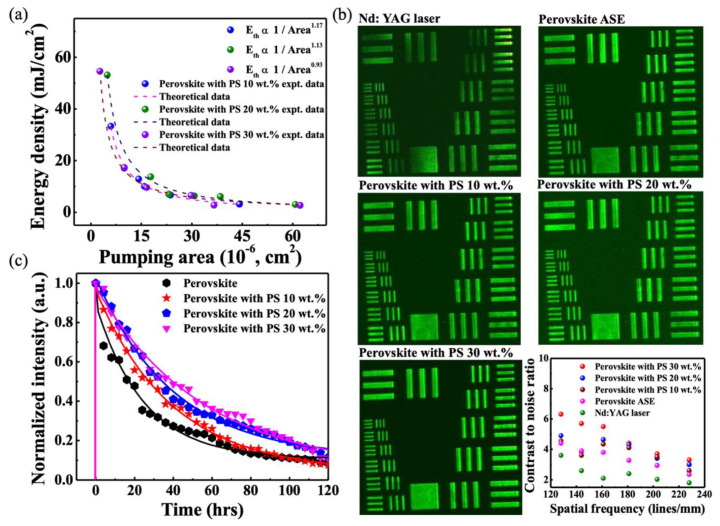
Characteristics and speckle free imaging using different random lasers: (**a**) excitation area dependent threshold energy density variation for different random lasers. The dotted line indicates the simulated results fitted to the experimental data; (**b**) Speckle free imaging of USAF resolution test chart using different random lasers. Evaluation of contrast to noise ratio of speckle free images for different random lasers; (**c**) Photon lifetime test for different random lasers.

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
