# Peer review of "Effect of Passivation Layer on the Thin Film Perovskite Random Lasers"

_materials, 2020, doi:10.3390/ma13102322_

Round 1
Reviewer 1 Report
The hybrid perovskite and PS combination appears to be very promising in the field of lasing. The work, therefore, has the potential to lead to a key publication, but after some significant revision.
Introduction. It should be introduced that the perovskite used is based on Br: why did you choose this? What advantages does it bring?
“Fabrication and Experiments” and “Experimental Section”. Why are these two sections divided? The first part of chapter 2 would go into the experimental section. The drafting creates confusion because the results are beginning to be discussed in line 89. The suggestion is to remove the chapter on "Lasing Performance Analysis" and put from line 89 onwards in the "Results and Discussion" chapter.
Conclusions. What concentration is suggested by the authors, i.e. which is the best? Why did they stop at 30%?
Besides, since annealing is performed after the deposition of the PS, what happens at the interface between perovskite and PS? Do the authors think there may be changes to the grain boundaries? Have they checked that there are no changes to the perovskite bulk?
How can it be said that the material is of high quality if no chemical or structural characterization is presented? It is essential that at least XRD is added, EDS and XPS would be welcome.
Author Response
Dear Reviewer,
We are grateful for your valuable time, comments, and suggestions.
Please find the list of changes in the attached response to the reviewer file.
All the suggestions and comments are addressed properly, and also highlighted in the revised manuscript file.

Reviewer 2 Report
The authors present the fabrication and characterization of a random laser, based on a perovskite thin film protected by a layer of polystyrene at different concentrations. The layer of polystyrene improves the performance of the random laser in terms of threshold and lifetime, compared to the simple perovskite film.
Overall, I think this topic is interesting and this paper can be published after some points are revised:
Lines 212-221. In the lasing experiment it is not clear the geometrical arrangement. From which directions the laser is focused on the sample and the signal collected? Is the geometrical arrangement relevant for observing lasing?
How has the excitation area been varied?
Perovskites are known to be very sensitive to humidity as mention in the introduction and to heating. Experiments have been conducted under ambient condition. Have the authors had indications that samples were degrading prior or during the measurements? Was it important to work with very fresh samples?
How does the performance of the samples reported in this paper compare with other perovskite-based random lasers? I would report some comparison with the literature.
In the formula at line 162 the term A/A0 should be in parenthesis.
Author Response

(The authors gave the same response as above.)

Reviewer 3 Report
In this paper, the authors proposed perovskite – polystyrene based random lasers which can be used in speckle free high resolution imaging applications. They find out that 30 wt% concentration of polystyrene result in the largest contrast to noise ratio. They also studied energy density, intensity and USAF resolution test chart of their random lasers. The topic is novel, writing is good and overall presentation is clear. Therefore, I would recommend this work for publication in Materials Journal.
Author Response
Dear Reviewer,
We are grateful to your valuable time, and decision towards reviewing our manuscript and recommending it for publication in Materials.

Reviewer 4 Report
The authors in the manuscript show and discuss the effect of changing the concentration of a passivation layer, polystyrene, on the lasing performance of the thin film Perovskite random laser. Although their data are sound, there are some comments that the authors must address:
1- In figure 2, I can see multiple peaks. It is not clear in the manuscript which peaks the authors are talking about. A peak fitting discussion and figure should be provided. Also, the peak wavelength must be plotted vs. pump energy density.
2- Temperature dependent emission should be provided to understand the multiple peak origins and shed some light on these multiple peaks.
3- Since polystyrene is a passivation layer, and to assess the performance of the RL, the authors should provide evidence of RL stability and reliablility of the data over a period of time.
4- Authors should demonstrate the working principle schematically for this RL given the structure discussed in the manuscript.
5- each of figures 1c and 1d must be in a separate figure with larger dimensions.
Author Response
Dear Reviewer,
We are grateful for your valuable time, comments, and suggestions.
Please find the list of changes, in the attached response to the reviewer file.
All the reviewer’s suggestions and comments are addressed properly and also highlighted in the Manuscript file.

Reviewer 5 Report
Dear Authors,
Please find attached the Review 1 Report.
Best regards,

Author Response

(The authors gave the same response as above.)

Round 2
Reviewer 1 Report
The manuscript is much more enjoyable. The issues of version 1 have been adequately addressed.
Reviewer 4 Report
The revised manuscript has shown better improved, making the manuscript acceptable for publication.